# Zonarol Protected Liver from Methionine- and Choline-Deficient Diet-Induced Nonalcoholic Fatty Liver Disease in a Mouse Model

**DOI:** 10.3390/nu13103455

**Published:** 2021-09-29

**Authors:** Jia Han, Xin Guo, Tomoyuki Koyama, Daichi Kawai, Jing Zhang, Reimon Yamaguchi, Xiaolei Zhou, Yoshiharu Motoo, Takumi Satoh, Sohsuke Yamada

**Affiliations:** 1Department of Pathology and Laboratory Medicine, Kanazawa Medical University, Ishikawa 920-0293, Japan; hanj227@kanazawa-med.ac.jp (J.H.); lovejing_86@163.com (J.Z.); sohsuke@kanazawa-med.ac.jp (S.Y.); 2Department of Medical Oncology, Kanazawa Medical University, Ishikawa 920-0293, Japan; motoo@kanazawa-med.ac.jp; 3Department of Pathology, Kanazawa Medical University Hospital, Ishikawa 920-0293, Japan; 4Laboratory of Nutraceuticals and Functional Foods Science, Graduate School of Marine Science and Technology, Tokyo 108-8477, Japan; 5131yu.da.18@gmail.com (T.K.); n58j87@bma.biglobe.ne.jp (D.K.); 5Department of Dermatology, Kanazawa Medical University, Ishikawa 920-0293, Japan; raymon-y@kanazawa-med.ac.jp; 6College of Bioscience & Bioengineering, Hebei University of Science and Technology, Shijiazhuang 050018, China; foxlei@live.cn; 7Department of Anti-Aging Food Research, School of Bioscience and Biotechnology, Tokyo University of Technology, Tokyo 192-0982, Japan; satotkm@stf.teu.ac.jp

**Keywords:** NAFLD, zonarol, Nrf2

## Abstract

Nonalcoholic fatty liver disease (NAFLD) is one of the most common liver diseases with no approved treatment. Zonarol, an extract from brown algae, has been proven to have anti-inflammatory and antioxidant effects. In this study, we investigated the role of zonarol in the progression of methionine- and choline-deficiency (MCD) diet-induced NAFLD in mice. After oral treatment with zonarol, a lighter body weight was observed in zonarol group (ZG) mice in comparison to control group (CG) mice. The NAFLD scores of ZG mice were lower than those of CG mice. Hepatic and serum lipid levels were also lower in ZG mice with the reduced expression of lipid metabolism-related factors. Furthermore, ZG mice showed less lipid deposition, less inflammatory cell infiltration and lower inflammatory cytokine levels in comparison to CG mice. Moreover, the numbers of 8-hydroxy-20-deoxyguanosine (8-OHdG)-positive hepatocytes and levels of hepatic and serum thiobarbituric acid reactive substances (TBARS) were significantly lower in comparison to CG mice. The expression levels of nuclear factor erythroid 2 related factor 2 (Nrf2), as well as its upstream and downstream molecules, changed in ZG mice. Zonarol could prevent the progression of NAFLD by decreasing inflammatory responses, oxidative stress and improving lipid metabolism. Meanwhile the Nrf2 pathway may play an important role in these effects.

## 1. Introduction

Non-alcoholic fatty liver disease (NAFLD) is reported to have a prevalence of around 20–40% in Western countries and around 7.9–43.3% in Eastern countries [1] and is characterized by a steatotic hepatocyte population exceeding 5% in the liver, without the consumption of alcohol [2]. NAFLD can lead to fatal disease [3]. Its global prevalence is approximately 6–30% [4], and the number of cases is increasing year by year.

The pathological process of NAFLD follows a series of complex processes that can be described as steatosis, inflammation and hepatocyte death, with the loss of the repair function [5]. Steatosis is caused by abnormal hepatic lipid metabolism, such as an increased uptake of lipids and lipogenesis and/or reduced secretion of lipids in the liver [6]. Sterol regulatory element-binding protein 1/2 (SREBP1/2) and peroxisome proliferator-activated receptor γ (PPARγ) are very important transcription factors in mediating this process [7]. Then, steatosis enhances the release of endogenous damage-associated molecular patterns (DAMPs) by injured hepatocytes. DAMPs can activate the transcription factor nuclear factor-kappa B (NF-κB) and irritate the immune cell recruitment (mainly macrophages and Kupffer cells), which upregulates the production of inflammatory cytokines, such as tumor necrosis factor-alpha (TNF-α) and interleukin-1β (IL-1β) [8], recruit or activate Kupffer cells/macrophages and further aggravate hepatocyte damage and steatohepatitis [9].

Meanwhile, abnormal lipid deposition in the liver causes lipotoxicity, which leads to mitochondrial dysfunction and endoplasmic reticulum stress, and then induces the production of reactive oxygen species (ROS), which leads to oxidative stress [10]. The increased ROS can further lead to necrotizing inflammation and fibrosis through lipid peroxidation induced by the activation of stellate cells [11]. The pathogenesis of NAFLD is considered to be a vicious cycle of steatosis, lipotoxicity and inflammation that produces complex histopathological and biochemical changes in the liver, ultimately leading to cirrhosis and even hepatocellular carcinoma [12]. Nuclear factor erythroid 2 related factor 2 (Nrf2), as a key transcription factor, is not only a central regulator of intracellular redox homeostasis but also one of the most important signaling molecules related to hepatic metabolism and NAFLD [13]. Activated Nrf2 can induce its downstream target genes heme oxygenase 1 (HO-1) and NAD(P)H dehydrogenase (quinone 1) (NQO1), which have antioxidant and anti-inflammatory functions, and which regulate metabolic syndrome [14,15].

However, no specific drugs for the treatment of NAFLD have been developed [16]. While several drugs have certainly shown some benefit in treating NAFLD patients, they have also caused significant adverse reactions; thus, the identification of novel therapeutic drugs remains an ongoing challenge [17]. Recently, extracts of terrestrial and marine plants have attracted attention in relation to NAFLD therapy [18]. Zonarol is an extract from brown algae, which is usually included in the diet of individuals living coastal areas [19]. Zonarol was first reported to have antifungal activity in the 1970s [20]. A previous study showed that zonarol inhibited inflammation and apoptosis without any evident side effects in a mouse model of dextran sulfate sodium-induced colitis [21]. Furthermore, it has also been reported that zonarol exerts antioxidant effects on nerve cells through the activated Nrf2/antioxidant response elements (ARE) pathway [22].

Based on the above studies, we hypothesized that zonarol can also inhibit inflammation and oxidative stress in NAFLD. Therefore, in the present study, we used a methionine- and choline-deficient (MCD) diet-induced NAFLD model [11,23] to investigate the role of zonarol in preventing the progression of NAFLD in mice. We observed the expression profile of genes related to inflammation, oxidative stress and lipid metabolism in the liver of MCD-induced NAFLD mice. Furthermore, after oral treatment with zonarol, we examined the histopathological changes in the liver and underlying mechanism in NAFLD mice, with our findings suggesting that zonarol can effectively improve NAFLD and related complications and may be beneficial for patients with NAFLD.

## 2. Materials and Methods

### 2.1. Extraction and Purification of Zonarol

As described in our last paper [21], the zonarol we used in the present experiments was roughly extracted from brown algae collected from an intertidal area in Japan. The fresh algae were dried and extracted with a five-fold volume of methanol for five days. The whole crude extraction process was performed at room temperature. After filtering twice, followed by evaporation and freeze-drying, a dark-green powder was obtained. The active components were separated by partitioning and column chromatography and further separated until a single compound was detected by high-performance liquid chromatography. The purity of extracted zonarol was over 99%. The structure of the purified compound was then determined according to the spectral data from nuclear magnetic resonance procedures [22].

### 2.2. Experimental Animals

Eight-week-old male C57BL/6 mice (40 in total, bought body weight: 21.8 ± 1.2 g) purchased from Sankyo Labo Service Corporation, Inc. (Toyama, Japan), were used in our experiments. They were maintained at a temperature of 21–26 °C with a 12-h light-dark cycle with *ad libitum* access to water. As described previously [24], the NAFLD mouse model was established by 3-week MCD+ high fat (HF) diet (60% fat; KBT Oriental Corporation, Saga, Japan). After that, the MCD diet was stopped, and an high fat (HF) diet (60% fat; KBT Oriental Corporation, Saga, Japan) was given to maintain a hyperlipidemic condition. When the HF diet was started, all mice were randomly separated into (CG; *n* = 15) and a zonarol group (ZG; *n* = 15). ZG mice were given zonarol at 20 mg/kg body weight orally once a day for 2 weeks, while CG mice received oral saline as a control. After two weeks of oral administration, all mice were anesthetized with an injection of ketamine-medetomidine and euthanized by exsanguination (Figure 1) without fasting, 24 h after the last gavage administration. Whole-blood samples were taken from axillary vessels and kept at room temperature for 2 h until blood coagulation. All blood cells were then removed by centrifugation, and serum samples were frozen at −80 °C in a freezer for use in further experiments. The livers were cut into pieces. Part of the sample was frozen in O.C.T. compound (Tissue-Tek^®^ O.C.T. Compound; Sakura Finetek Japan Co., Ltd., Tokyo, Japan) with liquid nitrogen for frozen sectioning, while other parts were kept frozen at −80 °C or fixed in 10% neutral-buffered formalin for use in further experiments.

### 2.3. Ethics

All protocols of our experiments were approved by the Ethics Committee of Animal Care and Experimentation, Kanazawa Medical University, Japan. The code of our project was 2020-58. It was registered on 1 April 2020. Our experiments were carried out under the Institutional Guidelines for Animal Experiments and the Law (no. 105) and Notification (no. 6) of the Japanese government. The number of experimental animals was kept to a minimum, and the suffering of the animals was minimized.

### 2.4. Histopathology

After fixation in 10% neutral-buffered formalin over 24 h, the same liver lobe of all mice was selected, embedded in paraffin, cut into sequential sections of 3–5 μm in thickness and subjected to Hematoxylin-Eosin (HE) staining and immunohistochemistry (IHC). The frozen sections were cut into 5- to 8-μm sequential sections and subjected to Oil Red O staining (Oil Red O Stain Kit; Polysciences, Inc., Warrington, PA, USA). All section images were captured with the Nano Zoomer Digital Pathology Virtual Slide Viewer software program (Hamamatsu Photonics Corp, Hamamatsu, Japan). As described in a previous study [11], we used the NAFLD score (steatosis score and ballooning score) to assess the condition of the mouse liver (Table 1).

### 2.5. IHC

IHC was carried out to assess the liver condition. An anti-mouse Galectin-3 (Mac-2) monoclonal antibody (1:1000; Cedarlane Laboratories Ltd., Burlington, ON, Canada) was used to evaluate inflammation. We also used the NAFLD score to assess the liver condition of the mice (Table 1). Then, the liver tissues were stained with a monoclonal mouse anti-human α-smooth muscle actin (α-SMA) antibody (1:1000; Dako Cytomation, Carpenteria, CA, USA). In addition, 8-OHdG antibody (a marker of oxidative stress) staining was performed (1:20; Japan Institute for the Control of Aging, Shizuoka, Japan). The details of all antibodies used are shown in the Appendix A and Section 2. As described previously [25], the numbers of positive cells were counted in 10 random high-power fields (×200) in each slide by observers.

### 2.6. Western Blotting

Protein samples were extracted from mouse liver samples using radioimmunoprecipitation assay (RIPA; Sigma-Aldrich Co. LLC, St. Louis, MO, USA) lysis buffer, which contained a protease inhibitor cocktail. The protein samples were then denatured with 2× Laemmli sample buffer (Bio-Rad Laboratories, Inc., Hercules, CA, USA) at 95 °C for 5 min. The denatured protein samples were loaded onto 12.5% sodium dodecyl sulfate-polyacrylamide gel electrophoresis (SDS-PAGE) gels for electrophoresis and transferred onto Immun-Blot PVDF membranes (Bio-Rad Laboratories, K.K., Tokyo, Japan). The membranes were then incubated overnight at 4 °C on a shaker with a diluted primary antibody (β-Actin, IL-1β, NF-κB, HO-1, protein kinase B (Akt), phosphor-Akt, phosphoinositide 3-kinase (PI3K), phosphor-PI3K, Cell Signaling Technology, Inc., Danvers, MA, USA, 1:1000; PPARγ, TNF-α, kelch-like ECH-associated protein 1 (keap1) Abcam PLC., 1:1000; SREBP-1/2; Bioss Inc., Woburn, MA, USA, 1:500; Nrf2, Gene Tex, Inc., Irvine, CA, USA, 1:500; NQO1, Santa Cruz, 1:500). After washing with TBS-T buffer, the membranes were incubated with the secondary antibody (Cell Signaling Technology, Inc., Danvers, MA, USA 1:1000) at room temperature for 1 h. After visualization using an enhanced chemiluminescence kit (Bio-Rad Laboratories, Inc., Hercules, CA, USA), the membranes were checked and exposed by a luminescent image analyzer (LAS400; Fujifilm, Japan). All details of the antibodies used are shown in the Appendix A and Section 2.

### 2.7. The Analysis of the Tissue Lipid Content

To analyze the lipid content of the serum and liver, commercial assay kits (Wako Pure Chemical Co., Osaka, Japan) were used. As described in our previous paper [11], approximately 30 mg of liver tissue was used for extraction. Liver lipids were extracted with chloroform-methanol (2/1 *v*/*v*) and dried with a vacuum centrifuge before being resolubilized in 2-propanol.

### 2.8. Quantitative Real-Time Polymerase Chain Reaction

Quantitative real-time polymerase chain reaction was used to analyze the gene expression in liver tissue. Total RNA was extracted from the liver with a ReliaPrep™ RNA Tissue Miniprep kit (Promega, Madison, WI, USA). Custom primers and a TaqMan probe (TaqMan probes Applied Biosystems, Warrington, UK) for the amplification reaction were purchased from Life Technologies. The relative expression levels were normalized to 18S ribosomal RNA and the fold-change in ZG mice was calculated in comparison to CG mice.

### 2.9. Measurement of the TBARS Levels

A thiobarbituric acid reactive substances (TBARS) assay kit (Cayman Chemical Company, Ann Arbor, MI, USA) was used to analyze the TBARS levels in mice. Approximately 25 mg of liver tissue from each sample was homogenized on ice in 250 μL RIPA buffer containing a protease inhibitor cocktail, and centrifuged at 1600× *g* for 10 min at 4 °C to obtain the supernatant. Twenty-five microliters of homogenized sample of liver tissue was added to a reaction mixture containing 25 μL of 8.1% (*w*/*v*) SDS, 375 μL of 20% (*v/v*) acetic acid, and 375 μL of 0.8% (*w*/*v*) thiobarbituric acid, and diluted with distilled water to 1 mL. Samples were then boiled for 1 h and immediately moved onto ice and were incubated for 10 min. The reaction mixture was then centrifuged at 4° C, 1600× *g* for 10 min. We measured the absorbance spectrophotometrically in 100 μL of supernatant at a wavelength of 530–540 nm.

### 2.10. Statistical Analyses

All of our research data were expressed as box and whiskers plots, with a minimum to maximum bar, which showed all data-points using the GraphPad Prism software program. Significant differences were analyzed using a two-sided Student’s *t*-test and chi-squared test. Two-sided Student’s *t*-tests were conducted using the Microsoft Excel software program. Chi-squared tests were performed using the EZR software program. *p* values of <0.05 were considered to indicate statistical significance.

## 3. Results

### 3.1. Gavage Administration of Zonarol Restrained the Body Weight Gain Induced by HF Diet Feeding, Meanwhile the Liver Condition Was Improved

After 3 weeks of MCD + HF diet feeding, severe NAFLD was induced in mice with alteration of the expression of a series of genes related to inflammation, lipid metabolism and oxidative stress (Appendix A). Then, after two weeks of oral treatment with zonarol, the body weight gain caused by HF diet feeding in the zonarol group (ZG) mice was found to be restrained in comparison to the control group (CG) mice (CG mice vs. ZG mice: 24.4 ± 1.3 g vs. 22.2 ± 0.8 g; *n* = 15, *p* < 0.01) (Figure 2A). Although there was no apparent change between the two groups. HE-stained sections showed that the hepatic injury induced by MCD diet feeding was improved in liver of ZG mice. Livers from ZG mice showed fewer lipid droplets and inflammatory foci in comparison to those from CG mice, which conversely showed ballooning hepatocyte degeneration and inflammation (Figure 2B left panel and Table 2). In addition, Oil Red O staining showed less and smaller lipid deposition in ZG mice in comparison to that in CG mice (Figure 2B right panel). Furthermore, the NAFLD score of the livers of ZG mice was significantly lower in comparison to those of CG mice (Table 2).

### 3.2. After Zonarol Gavage Administration, the Lipid Contents of the Liver and Serum of Mice Tended to Be Decreased, While the Lipid Metabolism Was Improved

After zonarol gavage treatment, the liver triglyceride (TG) and nonesterified fatty acid (NEFA) levels in ZG mice were markedly decreased in comparison to CG mice (TG: CG mice vs. ZG mice: 39.8 ± 6.1 mg/g vs. 21.0 ± 3.1 mg/g, *n* = 15, *p* < 0.001; NEFA: CG mice vs. ZG mice: 0.20 ± 0.04 mEq/g vs. 0.14 ± 0.01 mEq/g, *n* = 15, *p* < 0.001) (Figure 3A upper panel). However, the liver total cholesterol (T-Cho) levels of the ZG and CG mice did not differ to a statistically significant extent. The serum lipid contents in ZG mice were also significantly decreased in comparison to CG mice (TG: CG mice vs. ZG mice: 151.5 ± 33.2 mg/dL vs. 105.6 ± 40.1 mg/dL, *n* = 15, *p* < 0.01; NEFA: CG mice vs. ZG mice: 1.0 ± 0.16 mEq/L vs. 0.82 ± 0.16 mEq/L, *n* = 15, *p* < 0.05; T-Cho: CG mice vs. ZG mice: 89.6 ± 16.1 mg/dL vs. 74.7 ± 17.9 mg/dL, *n* = 15, *p* < 0.05) (Figure 3A bottom panel). In addition to the lipid contents, the levels of transcriptional proteins related to lipid metabolism were reduced after the oral administration of zonarol; for example, the SREBP1/2 and PPARγ levels were both reduced at the protein and mRNA levels after the oral administration of zonarol (Figure 3B,C).

### 3.3. Inflammatory Reaction Was Significantly Suppressed in NAFLD Mice with Gavage Administration of Zonarol

The IHC results showed that there were obviously fewer macrophages (Kupffer cells), which were determined by Mac-2 in the liver of ZG mice, in comparison to CG mice (CG mice vs. ZG mice: 8.03 ± 2.4 vs. 1.04 ± 1.04, *n* = 10, *p* < 0.001) (Figure 4A upper panel). In addition, the numbers of stellate cells and fibrogenesis activation, which were determined by α-SMA, were also decreased in the liver of ZG mice in comparison to the CG mice (CG mice vs. ZG mice: 45.1 ± 1.6 vs. 21.5 ± 0.8, *n* = 10, *p* < 0.001) (Figure 4A bottom panels). Correspondingly, the levels of inflammatory cytokines and proinflammatory transcription factors, such as TNF-α, IL-1β and NF-κB, were also lower in the liver of ZG mice—at both the protein and mRNA levels—in comparison to CG mice (Figure 4B,C).

### 3.4. Oral Administration of Zonarol Protected the Liver from Oxidative Stress While the Nrf2 Pathway Played an Important Role

8-hydroxy-20-deoxyguanosine (8-OHdG) was used as the oxidative stress marker to evaluate the oxidative stress with IHC. Fewer 8-OHdG-positive cells were observed in the liver of ZG mice in comparison to CG mice (CG mice vs. ZG mice: 337.2 ± 25.1 vs. 247.7 ± 13.7, *n* = 10, *p* < 0.001) (Figure 5A). Another oxidative stress marker was also used to assess the liver and serum oxidative stress condition. The liver malondialdehyde (MDA) level was significantly decreased in ZG mice in comparison to CG mice (MDA: CG mice vs. ZG mice: 18.0 ± 5.6 μM vs. 11.7 ± 1.6 μM, *n* = 15, *p* < 0.05) (Figure 5B left panel), and the same tendency was also found in ZG mouse serum (MDA: CG mice vs. ZG mice: 21.2 ± 5.5 μM vs. 4.0 ± 2.4 μM, *n* = 15, *p* < 0.001) (Figure 5B right panel). In addition, the level of a well-known key transcription factor, Nrf2, was increased in the liver of ZG mice in comparison to that of CG mice, and its downstream regulatory molecule HO-1 and NQO1 showed the same tendency. In contrast, its negative regulatory factor, keap1, showed the opposite trend. Furthermore, another Nrf2 regulator, PI3K/Akt showed their activation in the phosphorylation format (Figure 5C).

## 4. Discussion

In this study, our results showed that gavage administration of zonarol obviously promoted the histopathological improvement of NAFLD in an MCD-induced mouse model, including reduced inflammatory cell infiltration, hepatic lipid deposition and oxidative stress, suggesting that zonarol may have positive effects in the treatment of NAFLD.

The circulation is a source of lipids in hepatocytes; thus, a decrease in serum lipids may lead to reduced lipid deposition in hepatocytes [26]. In this study, the NEFA and TG levels in the liver were relatively normalized in ZG mice, with the improvement of hyperlipidemia. Meanwhile, after zonarol treatment, the expression of some key regulators—including SREBPs and PPARγ—that were responsible for de novo lipogenesis and lipid storage were also downregulated. All of these results suggested that zonarol may have a regulatory role in the lipid metabolism of hepatocytes, which could inhibit the lipid deposition in hepatocytes and improve the lipid metabolism. Furthermore, the reduced lipid deposition within hepatocytes may also be a reason for the relatively mildly inflammatory response in the liver [27].

The abnormal accumulation of lipids in hepatocytes leads to liver damage, which causes an inflammatory response [28]; this event is very important in the progression of NAFLD [29]. Zonarol was reported to prevent inflammation in a mouse model of dextran sulfate sodium-induced ulcerative colitis by suppressing inflammatory responses, particularly by downregulating macrophage activation [21]. Indeed, in our study, macrophage infiltration and Kupffer cell activation were obviously reduced in the liver after the administration of zonarol. In line with these results, the hepatic expression of NF-κB, IL-1β and TNF-α was lower in ZG mice than in CG mice. These results indicated that zonarol has an effective anti-inflammatory role, not only in the colon, but also in the liver. This contributed directly or indirectly to regulate the expression of proinflammatory transcription factors so as to restrain the further development of inflammation in liver.

Accumulated lipids in hepatocytes can also cause dysfunction of the mitochondria and endoplasmic reticulum, leading to cellular oxidative stress, which is associated with inflammation and cell damage [30]. In addition, the destruction of hepatocytes caused by the inflammatory reaction can also cause an oxidative stress reaction of hepatocytes [31]. However, after zonarol treatment, we found that the oxidative stress response in the liver was significantly reduced. It was previously reported that zonarol was able to protect cells from oxidative stress—because of the properties of electrophilic compounds—by activating Nrf2 pathway [22]. Therefore, it can be inferred that the administration of zonarol can reduce the oxidative stress response of hepatocytes in model mice through similar mechanisms to those that reduce the liver injury.

Indeed, the Nrf2 signaling pathway in the hepatocytes was activated in the mice after the administration of zonarol. It has been reported that the loss of Nrf2 causes mild steatosis to rapidly progress to non-alcoholic steatohepatitis (NASH) [32], indicating that Nrf2 has an important role in the progression of NAFLD. The activity of Nrf2 can be regulated by keap1-dependent and keap1-independent mechanisms. In keap1-independent regulation, phosphorylation of PI3K/Akt would also activate Nrf2 [33]. The activated Nrf2 would induce downstream targets, such as HO-1 and NQO1, to protect liver from series lipotoxicity, to accordingly protect hepatocytes from damage [15,34]. In the present study, the expression of keap1 at the protein level was downregulated after the administration of zonarol, meanwhile PI3K/Akt were phosphorylated, indicating that zonarol may activate Nrf2 authentically through both keap1-dependent and keap1-independent mechanisms. Meanwhile, Nrf2 and its downstream targets, including HO-1 and NQO1 were also upregulated, indicating that zonarol treatment can protect hepatocytes through the activation of Nrf2 (Figure 6).

Several limitations associated with the present study warrant mention. Firstly, in the present study, metabolic cages were not used, so the changes in consumed feed and excreted feces were not analyzed after the gavage administration of zonarol. Therefore, we need to resolve the effects of zonarol on mouse eating patterns and intestinal nutrient absorption in a future study, although in the present study we did not find apparent differences in food intake or defecation only by our daily routine observation or histological differences in the small intestine between CG mice and ZG mice (data not shown). Secondly, the administration of zonarol was only conducted for two weeks; thus, the long-term side effects of zonarol administration are still unclear. Thirdly, there are still some molecular mechanisms that have not been clarified in this study. For example, the activation of Akt (known to be an inducer of SREBPs) did not match the low expression of SREBPs, although the regulation of SREBP is a complex process [35]. We plan to clarify these mechanisms in future in vivo studies. Finally, despite this MCD-induced NAFLD model being a classic one, several differences from human NAFLD, particularly with regard to certain metabolic changes, have been reported [36]. Therefore, whether or not zonarol can take the same significant effect in cases of human NAFLD remains to be confirmed. It is necessary to repeatedly verify the beneficial effects of zonarol with a more suitable animal model, for example the Gubra-Amylin NASH diet model [37].

In summary, zonarol prevented the progression of MCD-induced NAFLD by inhibiting the inflammatory response and oxidative stress and regulating the lipid metabolism in mice, resulting in the improvement of hyperlipidemia and obesity. Activated Nrf2 signaling may play a crucial role in these effects. Zonarol, as an extract from brown algae may be useful for treating patients with NAFLD.

## Figures and Tables

**Figure 1 nutrients-13-03455-f001:**
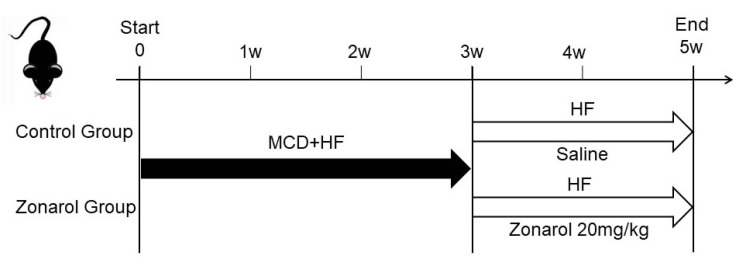
The NAFLD mouse model was established by feeding the mice with an MCD + HF diet for 3 weeks. Then, the mice were fed an HF diet to maintain their hyperlipidemic condition. When the high fat (HF) diet was started, all mice were randomly separated into the control group (CG) and a zonarol group (ZG). ZG mice were given zonarol orally at a dose of 20 mg/kg body weight once a day for 2 weeks, while CG mice were given saline orally as a control. MCD: methionine- and choline-deficient.

**Figure 2 nutrients-13-03455-f002:**
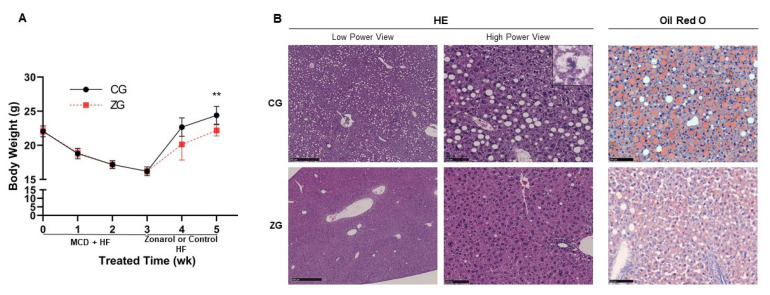
Gavage administration of zonarol inhibited the body weight gain caused by HF diet feeding and improved the liver condition in ZG mice. (**A**) Body weight changes in CG and ZG mice. (**B**) Representative micrographs of liver H&E staining (**left**) and Oil Red O staining (**right**) in HF and CG mice (low-power view: ×40, Bars = 500 μm; high-power view: ×200, Bars = 100 μm; *n* = 15 for H&E staining, *n* = 10 for Oil Red O staining). All data are presented as the mean ± standard deviation, ** *p* < 0.01; CG: control group, ZG: zonarol group.

**Figure 3 nutrients-13-03455-f003:**
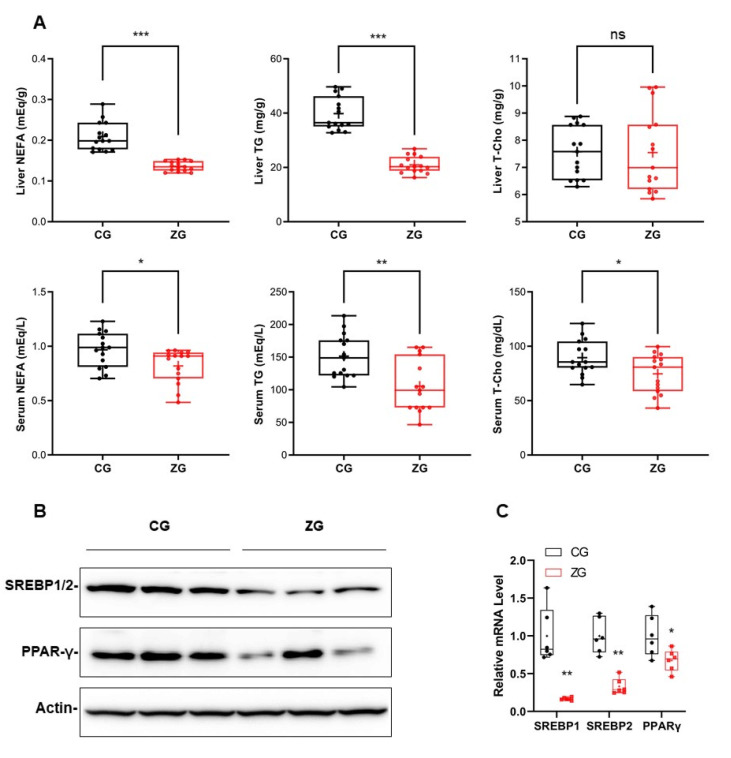
Gavage administration of zonarol improved the lipid metabolism in ZG mice. (**A**) Liver (**upper**) and serum (**bottom**) lipid levels. (**B**) Western blotting of liver transcription factor SREBP1/2 and PPARγ. (**C**) Quantitative real-time PCR of liver transcription factors SREBP1/2 and PPARγ. All data are presented as box and whiskers plots, with a minimum to maximum bar showing all data-points. The mean value showed as “+” in the box; * *p* < 0.05, ** *p* < 0.01, *** *p* < 0.001; CG: control group, ZG: zonarol group; TG, triglyceride; NEFA, nonesterified fatty acid; T-Cho, total cholesterol; SREBP1/2, sterol regulatory element-binding protein 1/2; PPARγ, peroxisome proliferator-activated receptor γ.

**Figure 4 nutrients-13-03455-f004:**
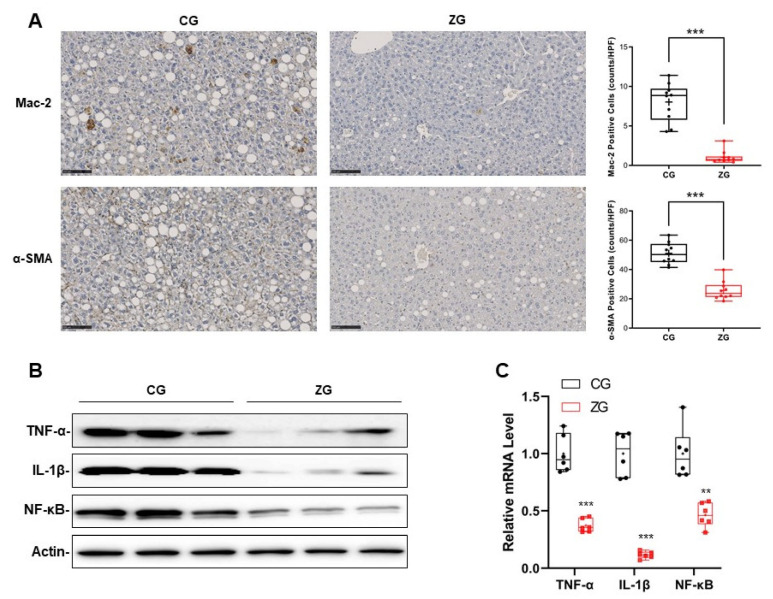
Gavage administration of zonarol suppressed the inflammatory reaction induced by MCD diet in ZG mice. (**A**) Immunocytochemistry to detect Mac-2 (**upper**) and α-SMA (**bottom**) in liver (×200, Bars = 100 μm, *n* = 10). (**B**) Western blotting of liver inflammatory cytokines TNF-α, IL-1β and transcription factor NF-κB. (**C**) Quantitative real-time PCR of liver inflammatory cytokines TNF-α, IL-1β and transcription factor NF-κB. All data are presented as box and whiskers plots, with the bar of the minimum to maximum values showing all data-points. The mean value is shown as “+” in the box; ** *p* < 0.01, *** *p* < 0.001. CG: control group, ZG: zonarol group.

**Figure 5 nutrients-13-03455-f005:**
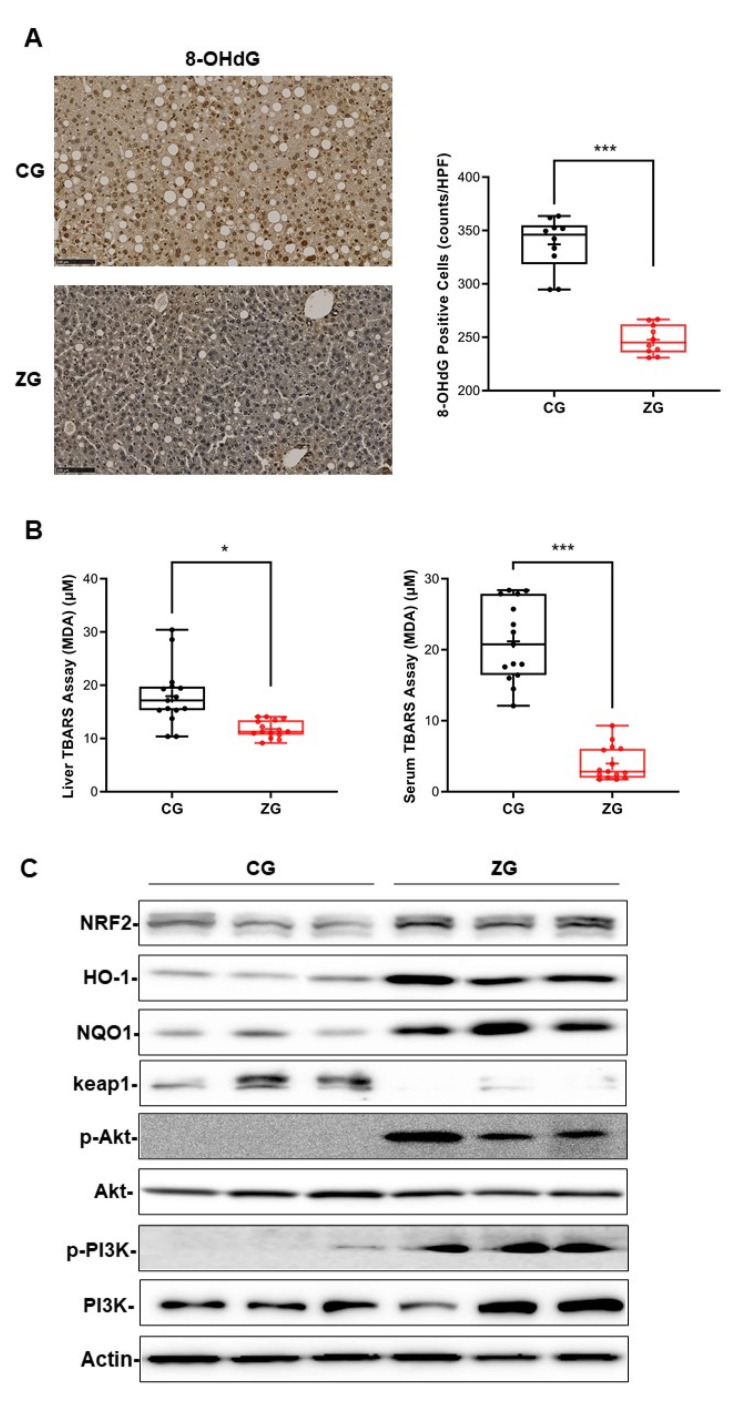
Gavage administration of prevented oxidative stress through the activation of Nrf2. (**A**) Immunocytochemistry (IHC) to detect 8-OHdG in liver (×200, Bars = 100 μm, *n* = 10). (**B**) Results of a thiobarbituric acid reactive substances assay of liver (**left**) and serum (**right**) (*n* = 15). (**C**) Western blotting of liver Nrf2, HO-1, NQO1, keap1 and PI3K, Akt with their phosphorylation formation. All data are presented as box and whiskers plots, with a minimum to maximum bar showing all data-points. The mean value showed as “+” in the box; * *p* < 0.05, *** *p* < 0.001. CG: control group, ZG: zonarol group; MCD, methionine- and choline-deficient.

**Figure 6 nutrients-13-03455-f006:**
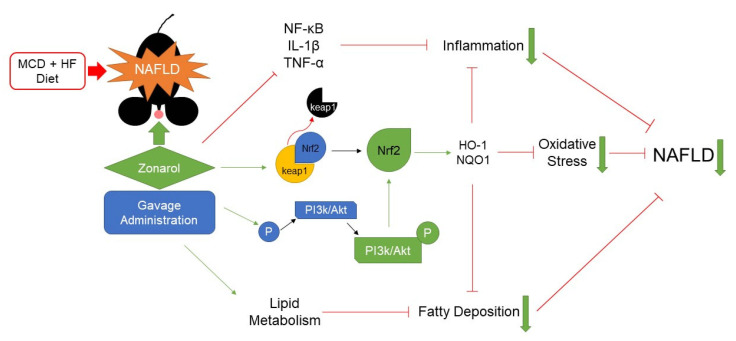
The administration of zonarol can improve the progression of NAFLD by inhibiting the inflammatory response and oxidative stress and regulating the lipid metabolism in mice, resulting in the improvement of hyperlipidemia and obesity through the activation of Nrf2. MCD: methionine- and choline-deficient; HF: high fat.

**Table 1 nutrients-13-03455-t001:** Quantitative NAFLD score for the grading of steatosis, inflammation and ballooning cells.

Steatosis Score		Inflammatory Score		Ballooning Score	
0	No lipid droplets.	0	No inflammation.	0	None.
1	Lipid droplets in <33% of hepatocytes.	1	<10 inflammatory foci, each consisting of >5 inflammatory cells.	1	Few ballooned cells.
2	Lipid droplets 33%–66% of hepatocytes.	2	≥10 inflammatory foci.	2	Many ballooned cells or prominent ballooning of cells
3	Lipid droplets in >66% of hepatocytes.	3	uncountable diffuse or fused inflammatory foci.		

**Table 2 nutrients-13-03455-t002:** Quantitative NAFLD score of steatosis, inflammation and ballooning cells in the liver.

Steatosis Score				Inflammation Score				Ballooning Score				NAFLD Score			
Score	CG	ZG	*p*	Score	CG	ZG	*p*	Score	CG	ZG	*p*	Score	CG	ZG	*p*
0	0	0	0.00904	0	0	4	0.00898	0	2	9	0.0155	0–3	1	9	0.00418
1	1	9		1	1	5		1	10	6		4–6	11	6	
2	8	4		2	12	6		2	3	0		7–8	3	0	
3	6	2		3	2	0									

*n* = 15. CG: control group, ZG: zonarol group.

## Data Availability

Not applicable.

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
