# Peer review of "Zonarol Protected Liver from Methionine- and Choline-Deficient Diet-Induced Nonalcoholic Fatty Liver Disease in a Mouse Model"

_nutrients, 2021, doi:10.3390/nu13103455_

Round 1

Reviewer 1 Report

Reviewer’s Comments:

This is an interesting study that illustrates the protective effects of Zonarol, an extraction from brown algae, in the progression of MCD diet-induced NAFLD. The authors revealed that Zonarol improves inflammatory responses, oxidative stress and lipid metabolism, through Nrf2 pathway in NAFLD model mice. However, before considering the manuscript for publication in Nutrients, the following points should be addressed.  

Major points

  1. Although data of Western blot, histology and RNA data showed statistically significant, bar graphs of data of liver NEFA, liver TG, serum NEFA, and serum TG, serum T-Cho in Figure 3 A do not seem significant at all. The reviewer would like to suggest an increase of Zonarol dose over 20 mg/kg.

  1. Although values described in text line 351 (CG mice vs. ZG mice: 337.2 ± 25.1 vs. 247.7 ± 13.7. n = 10, p < 0.005) seems significant, bar graphs shown in figure 5A do not seem significant at all.

There are discrepancies between experimental values and the figure graphs. The authors should fix the figure graphs correctly.

  1. Graphs of left panel of Figure 5B also do not seem significant at all.

Minor points (typographical errors):

  1. In line 277, period to comma.

Author Response

Response to reviewer#1:

First of all, we would like to thank reviewer for the helpful suggestions and comments. According to these suggestions and comments, we have revised our manuscript, especially the figures, to make them more distinct and clearer.

Major points

Although data of Western blot, histology and RNA data showed statistically significant, bar graphs of data of liver NEFA, liver TG, serum NEFA, and serum TG, serum T-Cho in Figure 3 A do not seem significant at all. The reviewer would like to suggest an increase of Zonarol dose over 20 mg/kg.

Response: Thank you for the suggestions and comments. Actually, before the formal experiment, we performed a pre-experiment using MCD diet NAFLD mice with different doses of zonarol, (20 mg/kg, 50 mg/kg and 100 mg/kg). Zonarol is not soluble in water or saline. It can only be dissolved in DMSO or methanol, which both are toxic to the body. In order to make a gavage solution that caused the least harm to mice as possible, we planned to dissolve zonarol in DMSO or methanol at concentrations of 40 mg/mL, 100 mg/mL and 200 mg/mL, then dilute with saline to reach a final concentration of 4mg/mL, 10 mg/mL and 20 mg/mL. So that each mouse in different groups would receive the same volume as 5 μL/g body weight with no more than 10% toxic solvent. However, zonarol could not be dissolved and dilute into a stable suspension when the concentration exceeded 4 mg/mL. Since the administration of over 200μL gavage at one time would place a great burden on the mice, we treated the 50 mg/kg and 100 mg/kg groups mice twice a day. However, the mice in the 50 mg/kg and 100 mg/kg groups soon showed gastrointestinal bleeding, accompanied by vomiting. Some could not eat normally. According to the principles of animal ethics, we terminated the experiment and obtained their liver samples. The histological results of the pre-experiment showed that doses of >20 mg/kg were of no benefit. Please see the histological figure in attachment.

Major points

Although values described in text line 351 (CG mice vs. ZG mice: 337.2 ± 25.1 vs. 247.7 ± 13.7. n = 10, p < 0.005) seems significant, bar graphs shown in figure 5A do not seem significant at all.

There are discrepancies between experimental values and the figure graphs. The authors should fix the figure graphs correctly.

Graphs of left panel of Figure 5B also do not seem significant at all.

Response: Thank you for pointing out this problem. We have already checked the raw data carefully and fixed all of the figure graphs. The figure graphs were modified into box and whiskers plots, with the bar of the minimum to maximum values showing all data-points. All plots were made using the GraphPad Prism software program. The revised parts of the manuscript are highlighted in yellow in the manuscript file. All data were once again tested by a two-sided Student's t-test and a simple ANOVA using Microsoft Excel and GraphPad Prism and a chi-squared test was performed using the EZR software program. P values of <0.05 were considered to indicate statistical significance. Please see the figure in attachment.

Minor points (typographical errors):

In line 277, period to comma.

Response: Thank you for your correction. We have corrected the manuscript file. After revising the manuscript, the manuscript was thoroughly checked by professional editor who is a native speaker of English (Dr. Brian Quinn from Japan Medical Communication).

Reviewer 2 Report

Han and coworkers tested whether Zonarol, a marine hydroquinone, prevents the progression of the NAFLD in methionine and choline deficient diet induced liver disease model.

 The authors successfully show that 20mg/kg gavage of Zonarol improved NAFLD score by histological measures, alpha SMA and Mac-2 staining, as well as by western blots and mRNA quantifications of inflammatory markers. Furthermore, the authors show strong correlation between Zonarol treatment and mRNA levels of SREBPs and PPAR gamma, suggesting that decrease lipid content in the livers might be an effect of transcriptional downregulation of de-novo lipogenesis.

Overall, the effects of Zonarol on the markers of NAFLD are striking and suggest that Zonarol could be used as an effective agent in the treatment of NAFLD in humans. Given the scarcity of available treatments for NAFLD, the discovery that Zonarol can be used to ameliorate NAFLD could be invaluable. However, there are major issues with the study design and the interpretation of results that the authors should address before it could be considered for publication.

Major issues.

  • While mice in Zonarol group have improved steatohepatitis and inflammation scores, there is no evidence whether it is mediated by the effect of Zonarol on the liver. The ZG mice gain less weight than the control group, and the authors did not comment on whether ZG mice eat the same amount of food as the CG mice. Decreased caloric intake and weight loss are known to ameliorate NAFLD, therefore it is critical to measure the food intake in both experimental groups to rule out that improved NAFLD is a result of decreased food intake. While the authors point out this and other shortcomings of the study design, without more rigorous investigation of Zonarol’s effects on food intake and excretion, as well as other related phenotypes such as insulin sensitive and glucose homeostasis, this study will only add confusion to the field.

  • MCD diet induced murine NAFLD/NASH model is not the best representation for human disease, and there is an alternative “non-deficient” diets available that more closely resemble the human diseases, both histologically and transcriptionally, such as Gubra-Amylin NASH (GAN) diet. Furthermore, there is no rational given for switching the diets in mid-study.

  • The authors should address the discrepancy in the histological (Oil Red O) and biochemical measurements of TGs in the livers: Oil Red O shows multiple fold decrease in lipid staining while liver TG measurements only show ~ 35% decrease.

  • The SREBP1/2 western blots are not informative and the authors should measure the mature form of SREBPs. Moreover, in their figure 5C the authors show complete absence of pAKT in their control group, while robust phosphorylation in ZG mice. This is accompanied with nearly 80% loss of SREBP1 mRNA which is counterintuitive as AKT is known inducer of SREBPs expression. Furthermore, it would have been more informative to measure some targets of PPARg and SREBP as well.
  • Incomplete methods section: no catalog numbers for any of the used antibodies, no mention of the state of animals at sacrifice, were they fasted? If yes, for how long? This is an important question when measuring liver triglycerides.

  • The discussion is poorly written and sometimes self-cites unrelated research. For example, in line 381-382 the authors state” The circulation is a source of lipids in hepatocytes, so a decrease in serum lipids may lead to reduced lipid deposition in hepatocytes” while citing a manuscript (11) that measured effects of acupuncture on progression of NAFLD.

Minor issues:

The graph bars should be represented with boxes that include individual datapoints.

The manuscript can hugely benefit from professional English editing services.  

Author Response

Response to reviewer#2:

At first, we would like to express our appreciation to the reviewer for his or her efforts to improve this manuscript. We have revised our manuscript, especially the Discussion, according to these suggestions and comments.

Major issues

While mice in Zonarol group have improved steatohepatitis and inflammation scores, there is no evidence whether it is mediated by the effect of Zonarol on the liver. The ZG mice gain less weight than the control group, and the authors did not comment on whether ZG mice eat the same amount of food as the CG mice. Decreased caloric intake and weight loss are known to ameliorate NAFLD, therefore it is critical to measure the food intake in both experimental groups to rule out that improved NAFLD is a result of decreased food intake. While the authors point out this and other shortcomings of the study design, without more rigorous investigation of Zonarol’s effects on food intake and excretion, as well as other related phenotypes such as insulin sensitive and glucose homeostasis, this study will only add confusion to the field.

Response: It is the one of the greatest limitations of this study that we did not measure eating and defecation of the mice after gavage administration. As indicated by the Reviewer, these are all critical aspects of our study. We would like to express our thanks to the Reviewer for giving us many valuable comments. We had already discussed these study limitations in a paragraph. Although we did not use the metabolic cage, we gave the mice feed quantitatively according to the general dietary law of mice, ignore the amount wasted. There was no obvious difference between the groups in terms of feed consumption and duration. Even the use of metabolic cages cannot completely avoid the waste of feed. In this study, we confirmed that Zonarol significantly inhibited inflammation and oxidative stress through Nrf2 related signaling pathway, which provides an important theoretical basis for its application in the treatment of fatty liver. Of course, it may also play a role in other aspects, such as insulin resistance, but a paper can't cover too much. Therefore, these aspects will be further verified by experiments in future research. We need to perform further experiments regarding the effects of zonarol on food intake and excretion as well. This is planned for our next study. Also, the reviewer suggested that we delete the confusing description about insulin sensitivity and glucose homeostasis. We have extensively revised one paragraph of the Discussion part. Please see the attachment.

Major issues

MCD diet induced murine NAFLD/NASH model is not the best representation for human disease, and there is an alternative “non-deficient” diets available that more closely resemble the human diseases, both histologically and transcriptionally, such as Gubra-Amylin NASH (GAN) diet. Furthermore, there is no rational given for switching the diets in mid-study.

Response: Thank you very much for your valuable suggestions. In this study, the MCD+ HF diet-induced NAFLD model was used to investigate the roles of zonarol in improving the progression of NAFLD. Even though some pathological manifestations are consistent between the two entities, the molecular mechanism underlying special diet-induced NAFLD does not totally reflect the pathogenesis of this disease in humans. Now that there are many dietary NAFLD models that are more in line with the characteristics of human NAFLD. It is possible to use them in future research to avoid the limitations of MCD+HF model. According to the suggestions, we have already revised the Discussion in response to the last suggestion. Besides, in the last two years, some papers have still described the use of the MCD NAFLD mouse model because it can quickly induce severe NAFLD.

Such as,

  1. Somm E, Montandon SA, Loizides-Mangold U et al. The GLP-1R agonist liraglutide limits hepatic lipotoxicity and inflammatory response in mice fed a methionine-choline deficient diet. Transl Res. 2021 Jan;227:75-88.
  2. Zhai T, Xu W, Liu Y et al. Honokiol Alleviates Methionine-Choline Deficient Diet-Induced Hepatic Steatosis and Oxidative Stress in C57BL/6 Mice by Regulating CFLAR-JNK Pathway. Oxid Med Cell Longev. 2020 Nov 27;2020:2313641.
  3. Vesković M, Labudović-Borović M, Mladenović D et al. Effect of Betaine Supplementation on Liver Tissue and Ultrastructural Changes in Methionine-Choline-Deficient Diet-Induced NAFLD. Microsc Microanal. 2020 Oct;26(5):997-1006.

Et al.

In this study, the mice were fed an MCD+ HF diet for 3 weeks followed by an HF diet for 2 weeks. One of the important reasons for switching the diets is that—in our group’s previous experiments—feeding with an MCD+HF diet for more than 3 weeks was found to lead to extreme weight loss of mice and death of some individuals, which is not in line with the provisions of experimental animal ethics. Therefore, we switched the diet to a 60% fat HF diet in order to ensure the development of hyperlipidemia.

Major issues

The authors should address the discrepancy in the histological (Oil Red O) and biochemical measurements of TGs in the livers: Oil Red O shows multiple fold decrease in lipid staining while liver TG measurements only show ~ 35% decrease.

Response: Thank you for your important comments. We have changed the ZG Oil Red O figure into a more general and representative field in the manuscript file. Please see the figure in attachment.

Major issues

The SREBP1/2 western blots are not informative and the authors should measure the mature form of SREBPs. Moreover, in their figure 5C the authors show complete absence of pAKT in their control group, while robust phosphorylation in ZG mice. This is accompanied with nearly 80% loss of SREBP1 mRNA which is counterintuitive as AKT is known inducer of SREBPs expression. Furthermore, it would have been more informative to measure some targets of PPARg and SREBP as well.

Response: Thank you for your valuable comments. PI3K/Akt is well known to regulate the expression of SREBPs, especially SREBP-1c. Meanwhile, the mechanism through which PI3K/Akt is involved in the regulation of SREBPs is very complex [Krycer JR, Sharpe LJ, Luu W, Brown AJ. The Akt-SREBP nexus: cell signaling meets lipid metabolism. Trends Endocrinol Metab. 2010 May;21(5):268-76.]. Rather than that, the data that we collected could support that zonarol protects against NAFLD via the activation of the PI3K/AKT/Nrf2 pathway. At the same time, some papers support the results of this study, for example [Jin M, Feng H, Wang Y et al. Gentiopicroside Ameliorates Oxidative Stress and Lipid Accumulation through Nuclear Factor Erythroid 2-Related Factor 2 Activation. Oxid Med Cell Longev. 2020 Jun 16;2020:2940746]. Nevertheless, PPARα was reported to downregulate SREBP-1c, which could improve NAFLD. At the same time, Nrf2 may also be involved in this process; this was not investigated in this study. In this study, we focused on the PI3K/AKT/Nrf2/ARE pathway, yet we plan to perform further research to address these issues in vivo using primary cultures of hepatocytes [Nabeshima A, Yamada S, Guo X et al. Peroxiredoxin 4 protects against nonalcoholic steatohepatitis and type 2 diabetes in a nongenetic mouse model. Antioxid Redox Signal. 2013 Dec 10;19(17):1983-98].

Major issues

Incomplete methods section: no catalog numbers for any of the used antibodies, no mention of the state of animals at sacrifice, were they fasted? If yes, for how long? This is an important question when measuring liver triglycerides.

Response: Thank you for the constructive comments. We have added a supplementary document recording all of the antibody catalog numbers used in this study. In both groups, no animals were not fasted and all animals were sacrificed 24 h after the last gavage administration. We tried our best to ensure that every animal was sacrificed under the same conditions. According to the suggestions, we have added these details to the Methods of the manuscript file.  Please see the attachment.

Major issues

The discussion is poorly written and sometimes self-cites unrelated research. For example, in line 381-382 the authors state” The circulation is a source of lipids in hepatocytes, so a decrease in serum lipids may lead to reduced lipid deposition in hepatocytes” while citing a manuscript (11) that measured effects of acupuncture on progression of NAFLD.

Response: Thank you for your precise comments. We revised the references here to another study that is more closely related to our research. We also improved the Discussion.

The circulation is a source of lipids in hepatocytes, so a decrease in serum lipids may lead to reduced lipid deposition in hepatocytes [25. Michele AB, David EC. Triglyceride Metabolism in the Liver. Compr Physiol. 2017 Dec 12;8(1):1-8.]

Minor issues

The graph bars should be represented with boxes that include individual datapoints.

Response: Thank you for your useful comments. We have already carefully revised all the figure graphs into box and whiskers plots with a bar of the minimum to maximum values, showing all data-points with the mean value indicated as “+” in the box. All plots were made using the GraphPad Prism software program. All data tested again by a two-sided Student's t-test and a simple ANOVA test using Microsoft Excel and GraphPad Prism and a chi-squared test by EZR software program. P values of <0.05 were considered to indicate statistical significance. Please see the attachment.

Minor issues

The manuscript can hugely benefit from professional English editing services.

Response: Thank you for your correction. We have corrected the manuscript file. After revising the manuscript, the manuscript was thoroughly checked by professional editor who is a native speaker of English (Dr. Brian Quinn from Japan Medical Communication).

Round 2

Reviewer 1 Report

The authors have fully answered the issues raised by the reviewer and made a considerable effort to clarify some critical aspects of the study, improving the description of some results. 

Author Response

Reviewer’s Comments:
The authors have fully answered the issues raised by the reviewer and made a considerable effort to clarify some critical aspects of the study, improving the description of some results.

Response: We would like to express our appreciation to the reviewer for his or her efforts to improve this manuscript. We have revised our manuscript according to these suggestions and comments and we requested a native English-speaking scientific doctor (Brian Quinn from Japan Medical Communication) to check the manuscript thoroughly. Thank you for your approvals.

Reviewer 2 Report

In this new revised version of the article, the authors corrected many shortcomings of the original manuscript. They should be commended for their efforts ( adding complete method sections, changing the figures to a more appropriate format, etc). However, the authors did not attempt to do any experiments, even simple ones that would not require additional animals. for example, new blots for mature SREBPs, quantification of Oil Red O, qPCR targets for SREBP and PPARg. Furthermore, in the discussing section, they change the text and now claim there did not find any apparent differences in food intake and defecation, while the original version says those parameters were not analyzed. 

My original concerns ( #1 and #4) are still not addressed. At least #1 concern should be addressed adequately.

Author Response

Reviewer’s Comments:
In this new revised version of the article, the authors corrected many shortcomings of the original manuscript. They should be commended for their efforts ( adding complete method sections, changing the figures to a more appropriate format, etc). However, the authors did not attempt to do any experiments, even simple ones that would not require additional animals. for example, new blots for mature SREBPs, quantification of Oil Red O, qPCR targets for SREBP and PPARg. Furthermore, in the discussing section, they change the text and now claim there did not find any apparent differences in food intake and defecation, while the original version says those parameters were not analyzed. 

My original concerns ( #1 and #4) are still not addressed. At least #1 concern should be addressed adequately.

Response: First of all, we would like to thank reviewer for the helpful suggestions and comments. According to these suggestions and comments, we have revised our manuscript according to these suggestions and comments.

We’d like to do experiments, but those additional experiments need time, especially the measurement of food intake and defecation in mice with metabolic cages. We need at least two months to complete these experiments, including the purchase of experimental materials, establishment of NAFLD model, gavage administration, the analysis of data and the revision of the manuscript. However, the last time, editor requested us to revise within 10 days, and this time within 3 days. We don't have enough time to do these experiments. Therefore, we plan to do these experiments in the future study.

We are sorry for the misunderstanding caused by the formulation in our manuscript. Actually, we did not find apparent differences in food intake or defecation only by our daily routine observation or histological differences in the small intestine between CG mice and ZG mice (data not shown). We didn't record the data, so we couldn't do statistical analysis.

Although we did not use the metabolic cage, we gave the mice feed quantitatively according to the general dietary law of mice, ignore the amount wasted, there was no significant difference between the groups in terms of feed consumption and duration. Even the use of metabolic cages cannot completely avoid the waste of feed. In this study, we confirmed that Zonarol significantly inhibited inflammation and oxidative stress through Nrf2 related signaling pathway, which provides an important theoretical basis for its application in the treatment of fatty liver. Of course, it may also play a role in other aspects, such as insulin resistance, but a paper can't cover too much. Therefore, these aspects will be further verified by experiments in future research.
